# Identification of Alcohol Use Prior to Major Cancer Surgery: Timeline Follow Back Interview Compared to Four Other Markers

**DOI:** 10.3390/cancers16122261

**Published:** 2024-06-18

**Authors:** Johanna Nicklasson, Moa Sjödell, Hanne Tønnesen, Susanne Vahr Lauridsen, Mette Rasmussen

**Affiliations:** 1Department of Health Sciences, Faculty of Medicine, Lund University, 22100 Lund, Sweden; johanna.nicklasson@skane.se (J.N.); moa.sjodell@skane.se (M.S.); 2WHO Collaborating Centre for Evidence-Based Clinical Health Promotion, The Parker Institute, Bispebjerg-Frederiksberg Hospital, University of Copenhagen, 2000 Copenhagen, Denmark; susanne.vahr.lauridsen.02@regionh.dk; 3Department of Surgery, Herlev-Gentofte Hospital, University of Copenhagen, 2730 Herlev, Denmark; 4Department of Clinical Medicine, University of Copenhagen, 2200 Copenhagen, Denmark; 5National Institute of Public Health, University of Southern Denmark, 1455 Copenhagen, Denmark; meras@sdu.dk

**Keywords:** alcohol drinking, cancer surgery, Timeline Follow Back, AUDIT–C, CDT, EtG, PEth, diagnostic, test performance, biomarkers

## Abstract

**Simple Summary:**

The postoperative complication rate is high (30–64%) among patients undergoing major surgery for bladder cancer, and significantly increased for patients with a high alcohol use at the time of surgical intervention. Several markers have been used to estimate alcohol use—including questionnaires and biomarkers. The aim of this study was to evaluate the accuracy for identifying preoperative alcohol use of four markers (one questionnaire and three biomarkers) relative to the standard method (Timeline Follow Back interviews). This was done in a sample of 94 patients scheduled for major bladder cancer surgery. We found none of the tested markers were sufficiently reliable to identify preoperative risky alcohol use and, for now, the standard procedure seems preferable.

**Abstract:**

Background: The postoperative complication rate is 30–64% among patients undergoing muscle-invasive and recurrent high-risk non-muscle-invasive bladder cancer surgery. Preoperative risky alcohol use increases the risk. The aim was to evaluate the accuracy of markers for identifying preoperative risky alcohol. Methods: Diagnostic test sub-study of a randomized controlled trial (STOP-OP trial), based on a cohort of 94 patients scheduled for major bladder cancer surgery. Identification of risky alcohol use using Timeline Follow Back interviews (TLFB) were compared to the AUDIT–C questionnaire and three biomarkers: carbohydrate-deficient transferrin in plasma (P–CDT), phosphatidyl-ethanol in blood (B–PEth), and ethyl glucuronide in urine (U–EtG). Results: The correlation between TLFB and AUDIT–C was strong (ρ = 0.75), while it was moderate between TLFB and the biomarkers (ρ = 0.55–0.65). Overall, sensitivity ranged from 56 to 82% and specificity from 38 to 100%. B–PEth showed the lowest sensitivity at 56%, but the highest specificity of 100%. All tests had high positive predictive values (79–100%), but low negative predictive values (42–55%). Conclusions: Despite high positive predictive values, negative predictive values were weak compared to TLFB. For now, TLFB interviews seem preferable for preoperative identification of risky alcohol use.

## 1. Introduction

Radical cystectomy (RC) with pelvic lymphadenectomy and urinary diversion is the standard surgical treatment for muscle-invasive bladder cancer and recurrent high-risk non-muscle-invasive bladder cancer. The complication rate is between 30 and 64% [1], and is significantly increased for patients with high alcohol use at the time of surgical intervention [2]. Drinking even a few drinks per day is associated with significantly increased postoperative morbidity of approximately 50% and approximately a 100% increase in mortality after surgery [2]. This surgical risk is in line with that originating from severely compromised cardiac and lung function. The impact of alcohol use in relation to the surgical outcome does not seem to differ among men and women and so far, about two drinks per day seems to be clinically accepted as the cut-off value. However, no safe use has been identified as reflected in the preoperative risk evaluation from the American Society of Anesthesiologists (ASA), which categorizes only persons with no or little alcohol use in the low-risk group [3,4].

The mechanisms behind this increased risk of postoperative complications include alcohol-initiated and subclinical dysfunction of several organ systems of major importance for the surgical intervention. Alcohol impairs the immune capacity, the cardiac function, the hemostatic balance as well as the stress response to the surgical intervention [5].

Still more evidence has been gathered about surgical risk reduction by quitting drinking at least four weeks prior to surgery [6]. The reduction in postoperative complications is based on the recovery from the organ dysfunctions during alcohol abstinence. However, a similar effect of reducing alcohol use remains to be demonstrated [7,8].

Identification of present alcohol use is a major challenge worldwide, despite the development of several tools to support identification in the clinical setting. The most common surgical routine is to ask the patients about their present daily or weekly use before the operation. Ascertaining present alcohol use can be improved by systematic use of the structured Timeline Follow Back (TLFB) model, covering use over the past 28 days [9]. A shorter timeline covering the past week only has been recommended for the surgical setting [10].

The questionnaire ‘alcohol use disorder identification test’ (AUDIT) in full or in the short form (AUDIT–C) is another tool used across healthcare sectors [11]. The AUDIT–C is a scaled marker and contains three questions on intake, frequency, and binge drinking over the last 12 months, however without the ability to reflect important changes like successful quitting four weeks prior to surgery [12].

Biomarkers for alcohol use have the advantage of being independent of recall bias, which may seem to be an attractive alternative. The most frequently used biomarker is the blood alcohol concentration, which is detectable only until the alcohol is metabolized. Other biomarkers include carbohydrate-deficient transferrin in plasma (P–CDT), phosphatidyl-ethanol in blood (B–PEth) [13], and ethyl glucuronide in urine (U–EtG) [14]. They have a longer detection window than blood alcohol concentration, but their different characteristics impact the interpretation and thereby the usability prior to surgery (Table 1).

P–CDT refers to sialic acid-deficient forms of transferrin formed in the liver. It is a clinically used biomarker for distinguishing an intake of 50–80 g ethanol per day for 1–2 weeks and levels return to normal in 2–3 weeks of abstinence. P–CDT levels differ with age, sex, and BMI [15]. P–CDT in plasma has a specificity of 70–100% but a wider range of 46–90% in sensitivity [17]. Cut off levels for long-time excessive drinking are indicated by levels >2% (P–CDT compared to total transferrin) [16].

B–PEth is a phospholipid formed in erythrocytes in the presence of ethanol, which clinically provides a very high sensitivity for detecting excessive alcohol use. It can be detected up to 2 weeks after a few days of high alcohol use [18]. Additionally, one-time consumption can be indicated as levels rise within 1–2 h [17]. Levels that exceed 0.30 µmol/L indicate excessive consumption of alcohol, whereas levels between 0.05 and 0.30 µmol/L correlate with moderate consumption and levels <0.05 µmol/L imply a non-existent or very low sporadic intake [16].

U–EtG is produced in small amounts and excreted through urine when ethanol is ingested and conjugated [20]. Sensitivity is high but dependent on quantity of ethanol use and diuresis [17]. U–EtG is fully eliminated within 30–70 h and is therefore a strong indicator of recent alcohol use [21]. A negative test, showing no recent alcohol consumption, is defined as ≤0.5 mg/L. Positive tests are confirmed with a mass spectrometric method [16].

As of today, the question of how to best identify patients at surgical risk due to risky alcohol use remains unanswered. To be able to offer a perioperative alcohol cessation intervention to relevant patients, the answer is highly relevant. The ideal marker to use in a surgical setting would be able to identify patients at surgical risk due to alcohol use and distinguish them from patients with a lower use. To evaluate the associations between the markers in this study, we have chosen an average of 2 drinks (24 g ethanol) per day, i.e., 14 drinks per week, preoperatively as the limit for identification [2].

The aim of this study was to evaluate the correlation between preoperative alcohol use identified by TLFB interviews and four other markers, AUDIT–C, P–CDT, B–PEth and U–EtG, in a cohort of patients scheduled for major bladder cancer surgery. Secondly, we wanted to evaluate the predictive values of the markers in addition to the sensitivity and specificity. We hypothesized that B–PEth would have the highest correlation and be the most reliable marker (see Table 1) to predict risky alcohol use.

## 2. Materials and Methods

### 2.1. Study Design

The project was performed as a cohort study with secondary analysis of data acquired in a randomized controlled trial, the STOP-OP trial, investigating the effect of an alcohol and/or smoking cessation intervention before major bladder cancer surgery on postoperative complications [23].

### 2.2. Study Population

This cohort consisted of 94 of 104 (90%) surgical patients in the original STOP-OP trial (eight did not have surgery and two withdrew consent shortly after randomization). The patients were scheduled for radical cystectomy at four Danish specialized urological centres with university affiliation, and they all underwent blood and urine sampling for biomarkers of alcohol use [23].

The inclusion criteria in the original STOP-OP trail were patients scheduled for radical cystectomy for bladder cancer, over the age of 18 years, smoking daily, and/or consuming at least three units of alcohol (36 g) daily. Patients were excluded if they were cognitively unable to provide informed consent, pregnant or breastfeeding, or allergic to Disulfiram, benzodiazepines, or Nicotine Replacement Therapy (NRT).

### 2.3. Outcomes

Information of alcohol use acquired through TLFB interviews [9], AUDIT–C [12], and three biomarkers, P–CDT, B–Peth, and U–EtG. All outcomes were measured prior to the operation.

### 2.4. Collection of Data

In this study, we used data collected at baseline. Alcohol consumption was collected by trained staff through TLFB interviews covering the past week. The intake was classified as standard drinks, each containing 12 g of ethanol. An AUDIT–C covering the past 12 months was also recorded.

The analyses for P–CDT (HPLC), B–PEth (Mass-Spectrometry and Electron Spray Ionization), and U–EtG (EMIT-test by Thermo Scientific^®^ (Waltham, MA, USA) analysed on Cobas 6000 c501 (Roche Diagnostic, Mannheim, Germany) and confirmation through Mass-Spectrometry) were performed blinded at the Department of Laboratory Medicine, Division of Clinical Chemistry and Pharmacology, Lund University, Lund, Sweden.

### 2.5. Ethics

The STOP-OP study was approved by the Danish Scientific Ethical Committee System (H-1-2013-134) and The Danish Data Protection Agency (2012-58-0004). The originally collected data included sensitive personal information, but all informed data used in this study were anonymized beforehand and handled in a secure manner to keep the patients’ integrity intact. All participants in the initial STOP-OP study gave consent including analysis of the alcohol biomarkers before data collection and randomization [23].

The STOP-OP trial was registered in Clinicaltrials.gov, Identifier: NCT02188446.

### 2.6. Statistical Analysis

The patient characteristics are presented in Table 2. In this study, we used non-parametric statistics as data and they were not assumed to be normally distributed. Correlations were examined using scatterplots to visualize and by calculation of Spearman’s rho correlation coefficient (ρ) to assess the monotonic associations. Correlations were interpreted as weak if ρ < 0.40, moderate if between 0.40 and <0.70, or strong if ρ ≥ 0.70 [24].

Sensitivity, specificity, predictive values, and receiver operation characteristic curves (ROC curves) including 95% confidence intervals (95% CIs) were used to evaluate the performance of AUDIT–C and the biomarkers. The 95% CI was estimated using a binomial distribution and the Wilson method was chosen as it is considered robust even for relatively small sample sizes, and in the case of a small number of successes or failures. For all statistical tests, *p* < 0.05 was considered significant. Statistical analyses were performed using R version 4.3.2 (R Foundation for Statistical Computing) [25], and the packages *caret* and *binom* for diagnostics, *pROC* for ROC curves, and *ggplot2* for graphs.

## 3. Results

All 94 patients in this study participated in the baseline interview and reported their alcohol use using both TLFB and AUDIT–C. Overall, 59 in 86 (69%) of the patients had at least one positive test result. Selected characteristics of the cohort are presented in Table 2.

### 3.1. Correlations

The correlation between the TLFB and AUDIT–C was strongest; ρ = 0.75 (ρ^2^ = 0.56). Regarding the three biomarkers, ρ was only moderate for all of them, ranging from 0.55 (ρ^2^ = 0.30) to 0.65 (ρ^2^ = 0.43), see Figure 1a–d.

### 3.2. Sensitivity, Specificity, and Predictive Values

Overall, the P–CDT reached the highest sensitivity at 82% but the lowest specificity at 38%. B–PEth showed the lowest sensitivity at 56%, but highest specificity of 100%. All tests had high positive predictive values, ranging from 79 to 100%, but their negative predictive values were low, 42–55%, see Table 3.

### 3.3. ROC Curves

In addition, ROC curves reflected the relative high sensitivity and specificity of all the alcohol tests, except for the low specificity of the P–CDT, see Figure 2a–d. AUDIT–C showed the highest area under the curve (AUC), indicating a high discrimination and prediction of the classification model. This was also the case for U–EtG and B–PEth.

## 4. Discussion

Overall, we found significant correlations between the structured patient interview on alcohol use identified by TLFB and all the other markers. All markers showed high positive prediction values, but weak negative prediction.

In the clinical setting, patients seldom overestimate their alcohol use. Therefore, the false positive self-reporting poses only a minimal problem. In contrast, underreporting is a general challenge, in both clinical and population-based studies [26] and false negative reporting constitutes a barrier for offering effective prehabilitation to reduce the surgical risk.

The significant correlations between the TLFB and the alcohol markers in our study support previous studies showing strong correlations, high sensitivity and specificity when using self-reporting [17]. Several different studies have used a variety of methods to assess alcohol use [27] resulting in a range of sensitivities and specificities, possible due to factors potentially influencing the formation and degradation of, e.g., PEth (i.e., hemoglobin, hematocrit, BMI, drinking pattern and rate, etc.) [28].

The weak negative predictive values in our study are surprising. They could be explained by a high prevalence of underestimating a high alcohol use at the TLFB interviews. However, this seems unlikely to be the main explanation as the patients included in the alcohol arms of this study reported an alcohol intake of at least 3 drinks per day or 21 drinks per week.

A more possible explanation of our results is that the different characteristics of each marker regarding the duration and amount of alcohol use would impact the associations (Table 1). For instance, the U–EtG is a strong indicator of any amount of present drinking, [21]; thus, inclusion of persons with a detected low alcohol use would lead to false negative results in our study group. Instead, this alcohol marker could be relevant for following up during intervention aiming at complete abstinence as in the 4-week prehabilitation [29] and the 6-month period recommended prior to liver transplantation [30].

AUDIT–C is another example of a sub-optimal alcohol marker for identification of preoperative alcohol use [11]. This test was developed for identification of alcohol use disorder based on the intake over the past 12 months. This poses a challenge in the surgical setting, where the required four-week successful quitting would not lead to a negative test result. This problem has been shown in a large cohort study on major surgery, which reported that the routinely obtained alcohol history close to the operation was better associated with postoperative complications than a positive AUDIT–C score [31]. To diminish the overestimation of risky alcohol use, preoperatively, a recent randomized trial used the AUDIT–C version, however, covering only the past 3-month period instead of the original 12 months [8].

In contrast, the P–CDT will be positive only at an intake of about 5 drinks per day for a few weeks and may therefore overlook a lower but still risky intake in relation to surgery [15].

We expected B–PEth to be the best biomarker to identify patients with a risky alcohol use prior to surgery because it will be positive when drinking at least 2 drinks per day or 14 per week, corresponding to the threshold used in this study. However, B–PEth also possessed a weak negative predictive value.

A main reason for using the TLFB interview in relation to cancer surgery and other types of surgery is that it has shown better results than even intensive data collection on daily reports. This may stem from interviewer assistance in counting larger or more concentrated drinks into standard drink sizes and encouragement for participants to use smartphones for more accurate recall of their activities [32].

### 4.1. Bias and Limitations

An overall challenge is that the characteristics of the different markers could be considered to fit better to some drinking patterns than to others, and therefore the characteristics of the study population may impact the results to a very high degree. A high and stable consumption over longer time may lead to positive identification for all the tests used in this study, while binge drinking or otherwise fluctuating use may lead to different results.

The small number of patients in our study may also impacts the result; likewise, the results should be interpreted with care as other populations, cultures, and regions may differ regarding average alcohol use, stigmatization, and access to free prehabilitation.

### 4.2. Perspectives

For the individual patient, it would have tremendous benefits to avoid postoperative complications by identifying and offering prehabilitation aimed at a high alcohol use prior to major cancer surgery. This requires training of the staff to offer effective prehabilitation in due time, reducing the postoperative complication rate by half [6].

From a clinical point of view, the fixed surgical agenda requires a high reliability of alcohol markers and questionnaires to distinguish between patients with a surgical risk from alcohol use (as well as to identify successful quitting over four weeks, preoperatively). On the other hand, it is also important not to misclassify the group of patients with a low alcohol use to be at surgical risk by using a suboptimal alcohol screening test, which would add stressful scenarios for the surgical patients, already physically and mentally impacted by the cancer disease.

Considering the societal perspective, it is meaningful to prevent potentially avoidable complications, as they incur significant costs due to their resource-intensive nature and thereby place a heavy burden on society at large.

Finally, it would be beneficial for the research area of cancer surgery and other surgical interventions to conduct a study enrolling a broader patient population, including patients with and without risk factors for increased postoperative complications in order to determine which marker is the most relevant for predicting complications in cancer surgery.

## 5. Conclusions

The negative predictive value of all three biomarkers and the AUDIT–C questionnaire compared to the TLFB interview seem too weak to be useful for preoperative identification of risky alcohol use, despite the high positive predictive values. For now, TLFB interviews seems preferable.

## Figures and Tables

**Figure 1 cancers-16-02261-f001:**
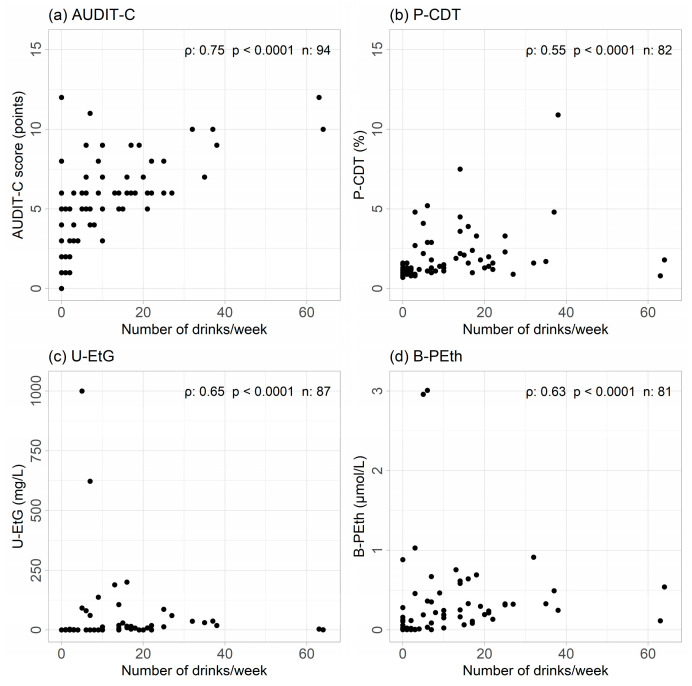
Correlations presented by ρ between alcohol use by Timeline Follow Back interview and (**a**) alcohol use disorder identification test (AUDIT–C); (**b**) carbo-deficient transferrin in plasma (P–CDT); (**c**) ethyl glucuronide in urine (U–EtG); (**d**) phosphatidyl-ethanol in blood (B–PEth).

**Figure 2 cancers-16-02261-f002:**
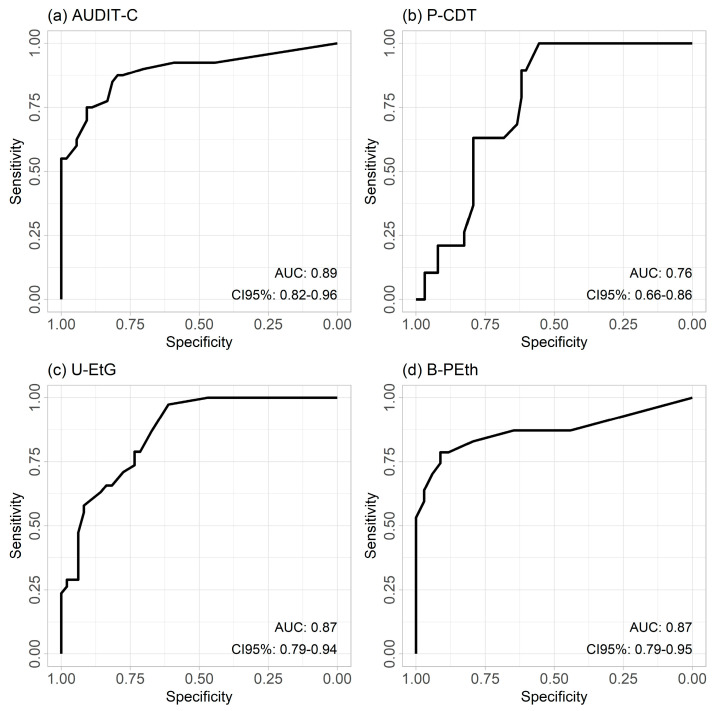
Receiver operating characteristic (ROC) curve for (**a**) AUDIT–C: alcohol use disorder-identification test, short version, and the biomarkers; (**b**) P–CDT: carbohydrate-deficient transferase in plasma; (**c**) U–EtG: ethyl glucuronide in urine; (**d**) B–PEth: phosphatidyl-ethanol in blood.

**Table 1 cancers-16-02261-t001:** Characteristics of the biomarkers frequently used to identify risky alcohol use.

Biomarkers	Carbo-Deficient Transferrin (CDT) [13,15,16,17]	Phosphatidyl-Ethanol (PEth) [16,17,18,19]	Ethyl Glucuronide (EtG) [16,17,20,21,22]
Half-life	2 weeks	4 days	2–3 h
Influenced by:			
Memory	-	-	-
Liver disease	+	-	-
Dehydration	+	-	Unknown
Diuresis	-	-	+
Blood transfusion	+	+	+
Intake for a positive result	~5 drinks/day for 2 weeks	~2–3 drinks/day for a few days	Any recent intake
Cut off value	>2% CDT/transferrin	≥0.050 µmol/L	>0.5 mg/L

**Table 2 cancers-16-02261-t002:** Characteristics of the 94 patients scheduled for major surgery for bladder cancer, presented as number (%) for categorical variables or median [range] for continuous variables.

Preoperative Characteristics	Values
Age (years)	67 [43–82]
Men	72 (77%)
Daily smokers	72 (77%)
Body–Mass Index (kg/m^2^)	25 [15–41]
Physical activity < ½ hour per day	33 (34%)
Living alone	42 (45%)
Education: none or only short courses	31 (33%)
**Alcohol characteristics**	
TLFB (drinks the last week)	4 [0–64]
≥21 units last week	13 (14%)
14–20 units last week	14 (15%)
1–13 units last week	40 (43%)
0 units last week	27 (29%)
AUDIT–C (points: 0–12)	5 [0–12]
P–CDT (% of total transferrin)	1.3 [0.7–10.9]
U–EtG (mg/L)	0.3 [0.0–1000]
B–PEth (µmol/L)	0.115 [0.003–3.010]
**History of disease**	
Tumor stage: cancer in situ	3 (3%)
Stage 1	31 (33%)
Stage 2	34 (36%)
Stage 3	21 (22%)
Stage 4	5 (5%)
Preoperative neoadjuvant chemotherapy	28 (30%)
Charlson Comorbidity Index ≥ 2	32 (34%)

Abbreviations: TLFB: Timeline Follow Back. AUDIT–C: Alcohol Use Disorders Identification Test, version C. P–CDT: Carbohydrate-deficient transferrin in plasma. U–EtG: Ethyl glucuronide in urine. B–PEth: Phosphatidyl-ethanol in blood.

**Table 3 cancers-16-02261-t003:** Sensitivity, specificity, and predictive values for identification of risky alcohol use among patients scheduled for major cancer surgery, compared to the use identified through Timeline Follow Back interviews and presented as % and 95% CI. (AUDIT–C: alcohol use disorder identification test, short version) and biomarkers (P–CDT: carbohydrate-deficient transferrin in plasma; B–PEth: phosphatidyl-ethanol in blood; U–EtG: ethyl glucuronide in urine).

Test	Positive Tests	Sensitivity		Specificity		Pos. Predictive Value		Neg. Predictive Value	
		%	95% CI	%	95% CI	%	95% CI	%	95% CI
AUDIT–C	40/94	75	(63–83)	96	(79–99)	98	(90–100)	55	(40–69)
P–CDT	19/82	82	(71–90)	38	(21–59)	79	(68–88)	42	(23–64)
U–EtG	38/87	70	(60–79)	86	(65–95)	94	(83–98)	47	(32–63)
B–PEth	47/81	56	(43–67)	100	(84–0)	100	(90–100)	43	(30–58)
Any test positive	59/86	43	(31–55)	100	(86–100)	100	(88–100)	40	(28–52)

## Data Availability

After study completion and publication of the study, the dataset analyzed during the current study will be available from the corresponding author on reasonable request.

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
