# Peer review of "Identification of Alcohol Use Prior to Major Cancer Surgery: Timeline Follow Back Interview Compared to Four Other Markers"

_cancers, 2024, doi:10.3390/cancers16122261_

Round 1
Reviewer 1 Report
Comments and Suggestions for Authors
In this study, the authors analyzed a cohort of 94 patients scheduled for major bladder surgery (i.e., radical cystectomy with pelvic lymphadenectomy and urinary diversion) for bladder cancer. This is a type of radical surgical intervention that is plagued by high postoperative complication rates, especially in patients with preoperative co-morbidities and major lifestyle risk factors. Among the latter, habitual alcohol consumption represents one major risk factor associated with significantly increased postoperative morbidity and mortality in this patient population. Therefore, a rigorous preoperative evaluation of alcohol consumption in this patient population is critical for improved postoperative success. This evaluation is presently done by subjecting these patients to the so-called structured timeline follow back (TLFB) interview, which evaluates the consumption of alcohol by these patients for about 1 month preoperatively. In the present study, the authors attempted to determine the performance of another questionnaire tool (i.e., the short form AUDIT-C) in combination with three biomarkers of alcohol use that can be measured in blood and urine (i.e., the measurement of the carbohydrate deficient transferrin in plasma (P-CDT), phosphatidyl-ethanol in blood (B-PEth), and ethyl glucuronide in urine (U-EtG)) for a more objective evaluation of alcohol use in these patients. The key findings of the study were a strong correlation between TLFB and AUDIT-C and a much more moderate one between TLFB and the three biomarkers of alcohol use. The statistical analysis of the collected data showed high positive predictive values for comparisons with TLFB but only weak negative predictive values for all tests. While the authors acknowledge the small sample size of the analyzed patient population as a major drawback of the study, they conclude that based on their findings TLFB interviews should remain the preferable tool for preoperative identification of risky alcohol use in this patient population.
The manuscript is well written and, based on the current methodology employed by the authors, the conclusions they draw are for the most part clear. However, I am not entirely convinced that the methodology used in this study is the most appropriate one and I think some additional clarifications are necessary. Therefore, I have a number of comments for the authors as follows.
1. My understanding is that the objective of the study was to determine the predictive performance of additional tools for evaluation of alcohol consumption (i.e., the AUDIT-C form and the three biomarkers) relative the standard interview tool (i.e., TLFB) which is used for this evaluation in the clinic. While this is generally fine, I am not entirely sure how practical this kind of comparison is. In other words, could the authors further comment on the translational value of their findings or how these findings may be applicable to clinical practice. In my opinion, it would be more practical to determine how these new tools for preoperative alcohol use evaluation actually perform against postoperative outcomes (i.e., the incidence and severity of postoperative complications). Unfortunately, the predictive values generated with the present study are not applicable to postoperative outcomes.
2. Related to the above, are the postoperative outcomes known for this patient cohort? Could they be added to the study to make additional correlations that would further better validate the performance of these new tools for preoperative alcohol use evaluation? In my opinion, the three-biomarker set employed in the study could be especially useful if further validated in relation to postoperative outcomes as objective predictive tools of preoperative alcohol use in this patient population. In this respect, it would be interesting to see how the B-Peth parameter (i.e., the most specific marker of alcohol consumption) would correlate with postoperative complications, either in isolation or together with the other two biomarker parameters.
3. The median value for P-CDT was 1.3% in this study which is under the reported cut-off value for this parameter (i.e., 2%). Therefore, I wonder how reliable is this number for at least half of the patient cohort enrolled in this study.
4. The values (range and median) for the B-PEth parameter that appear in Table 2 must be wrong. These values do not agree with the scale of the graph (panel d) for the same parameter which is shown in Figure 1.
Author Response
- My understanding is that the objective of the study was to determine the predictive performance of additional tools for evaluation of alcohol consumption (i.e., the AUDIT-C form and the three biomarkers) relative the standard interview tool (i.e., TLFB) which is used for this evaluation in the clinic. While this is generally fine, I am not entirely sure how practical this kind of comparison is. In other words, could the authors further comment on the translational value of their findings or how these findings may be applicable to clinical practice. In my opinion, it would be more practical to determine how these new tools for preoperative alcohol use evaluation actually perform against postoperative outcomes (i.e., the incidence and severity of postoperative complications). Unfortunately, the predictive values generated with the present study are not applicable to postoperative outcomes.
- Thank you for your suggestion. We agree that this would indeed be a very interesting study. However, in this study we only enrolled patients with risk factors (alcohol and/or smoking) that are known to increase the risk of postoperative complications.
It would be better to conduct such a study on a broader patient population, including both low- and high-risk patients. However, we have included the following in the perspectives:
“Finally, it would be beneficial for the research area of cancer surgery and other surgical interventions to conduct a study enrolling a broader patient population, including patients with and without risk factors for increased postoperative complications, in order to determine which marker is the most relevant for predicting complications in cancer surgery.”
- Related to the above, are the postoperative outcomes known for this patient cohort? Could they be added to the study to make additional correlations that would further better validate the performance of these new tools for preoperative alcohol use evaluation? In my opinion, the three-biomarker set employed in the study could be especially useful if further validated in relation to postoperative outcomes as objective predictive tools of preoperative alcohol use in this patient population. In this respect, it would be interesting to see how the B-Peth parameter (i.e., the most specific marker of alcohol consumption) would correlate with postoperative complications, either in isolation or together with the other two biomarker parameters.
- Again, thank you for your suggestion. Please see our answer to comment 1.
- The median value for P-CDT was 1.3% in this study which is under the reported cut-off value for this parameter (i.e., 2%). Therefore, I wonder how reliable is this number for at least half of the patient cohort enrolled in this study.
- It is true that the median P-CDT value in our study is well below the cut-off limit of 2% used to categorize the patients into positive and negative according to alcohol consumption. This only means that according to P-CDT, at least half of the patients are classified as negative (non-drinkers), which is in line with the result obtained using the TLFB interviews.
It is worth noticing that the P–CDT will be positive only at an intake of about 5 drinks per day for a few weeks and may therefore overlook a lower but still risky intake in relation to surgery [15].
- The values (range and median) for the B-PEth parameter that appear in Table 2 must be wrong. These values do not agree with the scale of the graph (panel d) for the same parameter which is shown in Figure 1.
- Thank you for noticing this. The CI should be [0.003-3.010] (and not [0.003-3,010]). This has now been changed.
Reviewer 2 Report
Comments and Suggestions for Authors
Reviewer Report
Identification of alcohol use prior to major cancer surgery: Timeline Follow-up interview compared to four other markers.
This is a well-described study that shows the improved identification of the biomarkers in the case of bladder cancer patients as a pre-operative precaution to avoid post-operative complications to the patients. The article needs to improve in some respects.
1. Can authors explain the entire study in the form of a graphical abstract?
2. What were the criteria used to calculate the standard drink of the patients please mention.
3. Are there any other potential bio-markers other than those mentioned in the study please list them out and explain briefly.
4. Please mention the gender of the patients
5. What were the exclusion criteria and inclusion criteria for the study any special other criteria used for recruiting the patients please mention?
Author Response
- Can authors explain the entire study in the form of a graphical abstract?
- We are aware of the possibility to present a graphical abstract, however, we have not been able to come up with an idea to explain the entire study in this form.
- What were the criteria used to calculate the standard drink of the patients please mention.
- As mentioned in section 4 Collection of data: “The intake was classified as standard drinks, each containing 12 g of ethanol.” This corresponds to the definition of a standard drink in Denmark.
- Are there any other potential bio-markers other than those mentioned in the study please list them out and explain briefly.
- We are aware that there are other bio-markers available, however, we have chosen to include only those relevant in a surgical setting.
- Please mention the gender of the patients
- 72 of the 94 of the patients were men. This was already listed in table 2.
- What were the exclusion criteria and inclusion criteria for the study any special other criteria used for recruiting the patients please mention?
- We have now added the following to section 2 Study population:
“The inclusion criteria in the original STOP-OP trail were patients scheduled for radical cystectomy for bladder cancer, over the age of 18 years, smoking daily and/or consuming at least three units of alcohol (36 g) daily. Patients were excluded if they were cognitively unable to provide informed consent, pregnant or breastfeeding, or allergic to Disulfiram, benzodiazepines, or Nicotine Replacement Therapy (NRT).”
Round 2
Reviewer 1 Report
Comments and Suggestions for Authors
I thank the authors for their responses to my comments. Their clarifications addressed my concerns in a satisfactory manner.